# Design and Structure of a Non-Coaxial Multi-Focal Composite Fresnel Acoustic Lens for Synergistic Ultrasound Stimulation of Multiple Brain Regions

**DOI:** 10.3390/s25113299

**Published:** 2025-05-24

**Authors:** Ruiqi Wu, Fangfang Shi, Juan Tao, Jiajia Zhao, Jinying Zhang, Xianmei Wu, Jingjing Xu

**Affiliations:** 1School of Integrated Circuits, Shandong University, Jinan 250101, China; 202200400065@mail.sdu.edu.cn (R.W.); 202332366@mail.sdu.edu.cn (J.T.); 202432446@mail.sdu.edu.cn (J.Z.); 2State Key Laboratory of Acoustics and Marine Information, Institute of Acoustics, Chinese Academy of Sciences, Beijing 100190, China; fangfangshi@mail.ioa.ac.cn; 3University of Chinese Academy of Sciences, Beijing 100049, China; 4Beijing Key Laboratory for Precision Optoelectronic Measurement Instrument and Technology, School of Optics and Photonics, Beijing Institute of Technology, Beijing 100081, China; jyzhang@bit.edu.cn

**Keywords:** transcranial focused ultrasound, Fresnel zone plates, non-coaxial focusing, acoustic lens

## Abstract

Transcranial focused ultrasound (TcFUS) neuromodulation is hindered by skull-induced acoustic limitations. To enable synergistic multi-region brain stimulation, we designed non-coaxial multi-focal composite Fresnel acoustic lenses, including an overlapping Fresnel lens (OFL) and an alternating-segmented Fresnel lens (ASFL). These lenses convert planar ultrasound into multiple foci. Based on Fresnel theory, acoustic fields were analyzed via simulations and experiments, validating the generation of four non-coaxial foci (10/30 mm focal lengths) from a 1 MHz planar wave using both OFL and ASFL. The influence of lens parameters on focal pressure distribution was investigated, and morphology was quantified using a linear least-squares method. Significant differences in focal morphology and intensity between OFL and ASFL provide crucial guidance for optimizing multi-target TcFUS strategies.

## 1. Introduction

Transcranial focused ultrasound (TcFUS) delivers focused acoustic energy to specific intracranial targets. Based on acoustic intensity, TcFUS encompasses high-intensity focused ultrasound (HIFU) for thermal ablation [1] and low-intensity focused ultrasound (LIFU). LIFU can modulate neural activity through mechanical effects [2] or permeabilize the blood–brain barrier via induced cavitation, often involving microbubbles [3,4]. Characterized by deep penetration [5], good directionality, and high spatial resolution [6], TcFUS demonstrates therapeutic potential for neurological disorders such as Alzheimer’s disease [3,7,8], epilepsy [9,10], and Parkinson’s disease [8,11].

However, skull heterogeneity in shape, structure, and material impedes precise TcFUS targeting by distorting the acoustic beam. Compensatory strategies have been developed. Phased-array focusing guided by intracranial hydrophones can correct for phase aberrations, but this invasive approach carries risks of brain damage [12]. Non-invasive phased-array techniques demand accurate patient-specific skull models, imposing significant computational loads, and exhibit high sensitivity to array configuration (position, element count [13]) and incident angle [14,15]. Acoustic time reversal, while self-adaptive in mitigating phase distortions, requires amplitude compensation in highly attenuating media like the skull, and its implementation in large-scale arrays remains challenging [16,17]. An alternative, more direct approach bypasses these compensation complexities by utilizing acoustic cranial windows. Alumina-based implants, for instance, can replace a portion of the skull to significantly reduce acoustic attenuation and phase distortion [18].

Furthermore, single-focus TcFUS often inadequately addresses neurological conditions spanning multiple brain regions, highlighting the need for multi-target strategies [19]. Brain functional complexity and overlapping neural networks frequently implicate multiple regions in both the pathology and effective treatment of disorders. For instance, bilateral subthalamic nucleus deep brain stimulation (DBS) demonstrates greater efficacy than unilateral DBS in Parkinson’s disease [20]. Similarly, concurrent multi-target ultrasound neuromodulation (USNM) of the prelimbic cortex and dorsal raphe nucleus in a depression mouse model yielded significantly enhanced antidepressant marker expression compared to single-target stimulation [21]. Therefore, synergistic multi-region stimulation offers the potential for more comprehensive target coverage, potentially reducing treatment sessions and improving overall therapeutic efficiency [22].

However, conventional single-element transducers are limited to a single intracranial focus and require high mechanical precision [23]. Phased arrays, while capable of multi-focal USNM, necessitate numerous elements, complex electronics, and meticulous intensity control to prevent adverse heating at targets and interfaces [24,25]. Acoustic lenses preceding single transducers can correct skull aberrations, but deriving their complex, often concave, profiles demands significant computational resources, and the geometry may risk transducer damage via air gaps [26,27].

Fresnel zone plates (FZPs), planar devices based on Fresnel half-wave band (FHWB) theory, offer focusing capabilities while circumventing the drawbacks of curved lenses. Bi-FZPs have been developed for dual-focal applications by combining two FZP designs [28]. However, these typically generate co-axial focal points, limiting their suitability for stimulating multiple, arbitrarily positioned brain regions simultaneously, a requirement for synergistic neuromodulation in complex disorders.

To address the need for low-intensity, multi-target synergistic TcFUS, this work introduces the design and structure of non-coaxial multi-focal composite Fresnel acoustic lenses. These lenses are engineered to generate multiple non-coaxial focal points in three dimensions for simultaneous neuromodulation of distinct brain regions. Specifically, we designed two configurations: an overlapping Fresnel lens (OFL) and an alternating-segmented Fresnel lens (ASFL). Their acoustic field characteristics were analyzed via finite element simulations and validated experimentally using an automated scanning system. Furthermore, the influence of key lens parameters on focal pressure distribution was investigated, and focal morphology was quantified using least-squares fitting. These findings provide critical theoretical and experimental guidance for optimizing multi-target TcFUS strategies.

## 2. Materials and Methods

### 2.1. Design Methodology of Composite Fresnel Acoustic Lens

As illustrated in Figure 1a, the lens design aimed to generate two sets of foci at a 1 MHz center frequency, targeting different depths. Within each set, foci were intended for bilateral stimulation, separated laterally by d=10 mm.

The principle for generating a single focus stems from the Fresnel half-wave band (FHWB) theory (Figure 1d). For a traditional Fresnel zone plate (FZP) focusing an incident plane acoustic wave, the radii of its zones adhere to the following relationship:(1)$$r_{n}=\sqrt{n\lambda F+{\left(\frac{n\lambda }{2}\right)}^{2}},$$
where $r_{n}$ is the radius of the *n*-th Fresnel zone, λ is the acoustic wavelength, F is the lens focal length, and n is the Fresnel zone number. In the specific design, all calculated Fresnel zone radii were arranged in an alternating pattern of inner and outer boundaries. For this study, the wavelength λ of 1 MHz ultrasound in water was used to compute the Fresnel zone radii at two focal lengths ($F_{1}=10\;\mathrm{mm}$ and $F_{2}=10\;\mathrm{mm}$). The Fresnel zone numbers n ranged from 1 to 18, with the calculated results presented in Table 1. Figure 1g illustrates the fundamental focusing principle: incident plane waves pass through the transmitting zones between the blocking material, converging at the target focus.

Next, to generate coaxial multi-foci, a coaxial bi-focal Fresnel zone plate (Bi-FZP) was designed by nesting an Inner FZP (for the $F_{1}=10\;\mathrm{mm}$ focus) and an Outer FZP (for the $F_{2}=30\;\mathrm{mm}$ focus), as depicted in Figure 1b,e. To ensure that the outermost radius of the Inner FZP is smaller than the innermost radius of the Outer FZP, while simultaneously aiming to maintain a comparable number of zones in both segments, the zone number ranges were selected as 1 to 11 for the Inner FZP and 5 to 18 for the Outer FZP in this study. As shown in Figure 1h, plane waves traversing the Inner FZP’s transmitting zones primarily contribute to the $F_{1}$ focus, while those traversing the Outer FZP’s transmitting zones contribute to the $F_{2}$ focus.

Subsequently, for bilateral stimulation, OFL was constructed by radially shifting an identical Bi-FZP by d=10 mm and superimposing it onto the original Bi-FZP in Figure 1c,f. In the OFL configuration (Figure 1i), incident plane waves transmit only through the overlapping transmitting zones of the two constituent Bi-FZPs. This geometric arrangement facilitates coherent superposition of the transmitted waves at four distinct spatial locations, thereby forming the four desired foci. Each underlying Bi-FZP unit contributes to generating its respective dual foci according to Fresnel principles.

The acoustic intensity distribution produced by the OFL can be calculated numerically using the Rayleigh–Sommerfeld diffraction integral. In a three-dimensional coordinate system, this integral is expressed as(2)$$p\left(x,y,z\right)=\frac{1}{j\lambda }\iint p_{i}\left(x^{\prime },y^{\prime }\right)\tau_{i}\left(x^{\prime },y^{\prime }\right)\frac{e^{-jkR_{1}}}{R_{1}}\cos\left(\theta \right)dx^{\prime }dy^{\prime },$$
where $p\left(x,y,z\right)$ represents the acoustic pressure at the target point $\left(x,y,z\right)$, λ is the ultrasound wavelength, $R_{1}=\sqrt{{\left(x-x^{\prime }\right)}^{2}+{\left(y-y^{\prime }\right)}^{2}+z^{2}}$, and $\cos\left(\theta \right)=z/R_{1}$. $p_{i}\left(x’,y’\right)$ denotes the complex pressure distribution of the incident field at the lens plane, and $\tau_{i}\left(x’,y’\right)$ is the transmission function of the composite Fresnel acoustic lens. It describes the change in complex amplitude (both magnitude and phase) of the wave passing through the lens at $\left(x’,y’\right)$, and is determined by the acoustic properties (impedance, sound speed, attenuation) and thickness of the lens material at that point. For this study, the blocking zones were fabricated from titanium alloy (TC4), and the transmitting zones consisted of polydimethylsiloxane (PDMS).

### 2.2. Numerical Method

The transmitted acoustic pressure field was calculated using the “Pressure Acoustics, Frequency Domain” interface within COMSOL Multiphysics^®^ version 6.2 (COMSOL AB, Stockholm, Sweden). Due to the non-axisymmetric nature of the designed lenses, a full 3D model was employed for simulation and analysis.

Based on the method of combining the constituent Bi-FZPs, the multi-focal Fresnel lenses are categorized into OFL and ASFL. The OFL structure (Figure 2a) was derived from a single Bi-FZP element by axially translating it by d=10 mm to obtain two copies, and constructing the composite by taking the spatial union of their original blocking zones. This resulted in an unequal area reduction in the Inner and Outer FZP transmitting zones, causing a significant intensity difference between the far-field and near-field focal spots. To address this, the ASFL structure (Figure 2b) was proposed. Unlike the OFL, the ASFL design, also based on superimposing two axially translated Bi-FZP copies, features the removal of the overlapping blocking zone areas. This approach ensured a more balanced area reduction across the Inner and Outer FZP transmitting zones, leading to improved focal spot intensity uniformity. The ASFL further includes 2 mm-wide rectangular supports, angled at 90° on each side and matching the blocking zone thickness, to provide mechanical stability and structural strength.

Figure 2c shows the 3D simulation model. In this setup, a planar transducer (D=50 mm diameter) was positioned 5 mm from the lens and driven with ultrasound excitation at a 1 MHz center frequency. Analyses were performed for lens thicknesses $t_{h}$ of 1 mm and 1.5 mm. The model boundaries were enclosed by perfectly matched layers (PML) to prevent interference from boundary reflections affecting the simulation results.

The model geometry was discretized using a free triangular mesh. Specific material parameters employed in the simulation are detailed in Table 2.

Solving the Helmholtz equation for each node of the discretized model,(3)$$\nabla \left(-\frac{1}{\rho_{0}}\left(\nabla p\right)\right)=\frac{\omega^{2}p}{\rho_{0}c^{2}},$$
yields the acoustic field distribution after ultrasound transmission through the composite Fresnel acoustic lens. Here, $\rho_{0}$ represents the medium density, c is the speed of sound, ω is the angular frequency, and p denotes the acoustic pressure.

### 2.3. Experimental Setup

As depicted in Figure 3a,b, the acoustic lens border structures were fabricated via 3D printing using direct metal laser melting (DMLM) with TC4 titanium alloy (Ti-6Al-4V, Ti Gr5; 3D Systems, Rock Hill, SC, USA). The transmitting zones were filled with polydimethylsiloxane (PDMS, Sylgard^TM^ 184, Dow Inc., Midland, MI, USA), prepared by mixing the base and curing agent at a 10:1 mass ratio followed by curing.

Acoustic field measurements were performed using a UPK-T36 automated C-scan system (Physical Acoustics Corporation, Princeton Junction, NJ, USA), equipped with an ADIPR-EXPRESS module (Figure 3c). This system enabled high-precision 3D acoustic field mapping within a scanning range of 900 × 600 × 450 mm. A 1 MHz-center-frequency KDYW-1M-01G planar piston transducer (Hangzhou Umbrella Automation Technology Co., Ltd., Hangzhou, China) served as the ultrasound source. Acoustic field signals were detected using a needle hydrophone with a sensitivity of 11 nV/Pa.

To ensure precise alignment and coaxiality between the lens and transducer, a custom-designed lens fixture was utilized. Fabricated through stereolithography (SLA) 3D printing using photosensitive resin, this fixture incorporated a top-mounted spirit level for precision leveling, guaranteeing parallelism between the lens plane and the transducer emitting surface. The fixture was mounted onto an optical platform via M6 screws at its base. To mitigate interference in the focal zone acoustic field measurements caused by multiple reflections between the transducer and the lens front surface, the separation distance (L) was maintained at 250 mm.

The ultrasound transducer was driven by a 1 MHz tone burst signal, generated by an arbitrary waveform generator and amplified by a power amplifier. This signal comprised 15 cycles of a sine wave, selected to ensure sufficient coherent summation of acoustic waves propagating from both the lens center and periphery to the intended focal position. A peak excitation voltage of 100 V was applied, with the driving source exhibiting an output impedance of 100 Ω.

Signals received by the hydrophone were processed. Processing parameters included a gain of 60 dB, a sampling rate of 50 MHz, and effective filtering of out-of-band noise using a bandpass filter with a passband of 0.4–3 MHz.

Three-dimensional acoustic field mapping was performed in a deionized water tank. The UPK-T36 automated C-scan system precisely controlled the movement of the hydrophone along the X, Y, and Z axes. The resolution for both scanning and stepping axes was set to 0.2 mm, and the movement speed was 40 mm/s. The system automatically recorded the detected sound pressure amplitude and phase information at each measurement point. For each lens configuration, the acoustic field was scanned in the XY plane at the lens center and YZ planes located at distances of 10 mm and 30 mm from the lens surface to fully characterize the morphology of each focal spot.

## 3. Results and Discussion

Figure 4b (sim.) and Figure 4c (exp.) illustrate the morphology of the four focal spots formed in the XY plane after a 1 MHz ultrasound signal traversed the OFL. These results confirm that the OFL structure effectively generates four distinct focal regions. In simulation, the peak pressure coordinates were found at (4.98, −11.18) mm, (−4.98, −10.95) mm, (4.98, −29.72) mm, and (−4.98, −29.72) mm, closely approximating the target positions. Experimentally, these peaks were measured at (4.9, −10.4) mm, (−4.9, −10) mm, (4.9, −29.6) mm, and (−5.1, −28.4) mm, showing good overall agreement with simulations. Notably, the experimental center of the left far-filed focus zone exhibited a relatively larger deviation from its target position, potentially attributable to fabrication inaccuracies inherent in the 3D printing process.

Figure 4a (sim.), Figure 4d (exp.) ($F_{1}=10\;\mathrm{mm}$) and Figure 4e (sim.), Figure 4f (exp.) ($F_{2}=30\;\mathrm{mm}$) present the acoustic pressure distributions in the ZY focal plane after 1 MHz plane waves traversed the OFL. The corresponding pressure profiles along the line connecting these foci are displayed in Figure 5a ($F_{1}=10\;\mathrm{mm}$) and Figure 5c ($F_{2}=30\;\mathrm{mm}$). Results indicate that both simulated and experimental main focal peaks align well with the target locations. However, in the OFL experimental results at $F_{2}=30\;\mathrm{mm}$ depth, significant central side lobes flanking both the left and right primary focal peaks were observed, reaching approximately 50% of the main peak intensity. Such strong side lobes were notably absent in the corresponding simulation. This discrepancy likely arises from wavefront aberrations in the practical incident wave, deviating from the ideal plane wave assumption, particularly near the lens periphery. The Outer FZP region, responsible for the far-field focus spots and situated at the lens exterior, is inherently more sensitive to such incident field non-uniformities.

Regarding ASFL, Figure 4h presents that this structure also effectively generated four distinct focal regions. The simulated peak pressure coordinates were located at (4.98, −11.40) mm, (−4.98, −11.63) mm, (4.98, −29.49) mm, and (−4.98, −29.94) mm. The ASFL simulation exhibited significant side lobes along the X-direction, distinct from the main foci. This artifact likely arises from the incomplete nature of the Fresnel zones in the ASFL design, allowing some transmitted acoustic energy to diffract and interfere outside the intended focal plane, thus creating undesired side lobes.

In the experimental results shown in Figure 4i, the peak pressure coordinates were measured at (4.8, −10) mm, (−4.8, −10.6) mm, (5.2, −25) mm, and (−4.4, −29.2) mm. Deviations between the measured and simulated focal positions were observed, potentially attributable to non-uniformity in the incident acoustic field during experiments. Notably, the deviation for the far-field foci was more pronounced for the ASFL relative to simulations. This finding suggests that the ASFL structure is more sensitive to the directionality and uniformity of the incident acoustic field compared to the OFL.

Figure 4g (sim.), Figure 4j (exp.) and Figure 4k (sim.), Figure 4l (exp.) show the focal spot morphologies at $F_{1}=10\;\mathrm{mm}$ and $F_{2}=30\;\mathrm{mm}$ in the ZY plane through the ASFL, respectively. Figure 5b ($F_{1}=10\;\mathrm{mm}$) and Figure 5d ($F_{2}=30\;\mathrm{mm}$) display the corresponding acoustic pressure profiles along the focal axes. Compared to the OFL, ASFL exhibits higher background acoustic pressure between the twin peaks at the $F_{1}=10\;\mathrm{mm}$ depth. Additionally, more pronounced side lobes are observed in peripheral regions far from the center. These characteristics are also likely attributable to the incomplete Fresnel zone structure inherent in the ASFL design.

To evaluate the acoustic transmission characteristics of the different lens designs, Figure 6a (sim.) and Figure 6b (exp.) compare the normalized acoustic pressure distributions along the X-axis for OFL and ASFL structures with thicknesses of 1 mm and 1.5 mm. Each profile represents the average pressure of the corresponding left and right foci, normalized to the maximum pressure within its respective dataset. Good agreement is observed between experimental and simulation results regarding main focal spot locations and side lobe features.

The results indicate that acoustic pressure in the near-field focal zone is significantly higher than in the far-field focal zone. This is attributed to the effective incident acoustic field area being smaller than the total lens area, resulting in less energy passing through the Outer FZP (responsible for the far-field foci) compared to the Inner FZP (near-field foci). For a given lens type, the peak focal pressure is higher for the 1 mm-thick lens compared to the 1.5 mm version, primarily due to lower ultrasonic attenuation in the thinner material. At the same thickness, the ASFL structure yields notably higher near-field focal pressures than the OFL, owing to its larger effective acoustic transmission area.

Additionally, side lobe structures were observed near X ≈ 17 mm and X ≈ 39 mm, beyond the primary foci, in both simulations and experiments, with relatively higher intensity for the ASFL compared to the OFL. The formation of these side lobes is attributable to constructive interference at specific locations between acoustic waves originating from the nested inner and outer Fresnel zones. This interference effect is more pronounced in the ASFL due to the incompleteness of its Fresnel zone structure, resulting in stronger side lobes.

Based on numerical simulations and experimental validation, we conducted detailed quantitative characterization of the designed planar acoustic lenses to demonstrate their capability for precise focal spot control. In terms of focal spot positioning accuracy, experimental measurements showed average axial deviations of 0.6 mm for OFL and 1.5 mm for ASFL, with corresponding average lateral deviations of 0.1 mm and 0.3 mm, demonstrating the excellent focal spot targeting capability of the designed lenses. Spatial resolution was quantified by the −3 dB acoustic field width of the focal spot. Experimental measurements revealed average lateral −3 dB acoustic field widths ranging from 1.47 mm to 1.81 mm for both ASFL and OFL structures across 10 mm and 30 mm depths. These values are comparable to the working wavelength of 1 MHz ultrasound in water, indicating high spatial resolution focusing. The corresponding average axial −3 dB lengths ranged from 3.01 mm to 8.47 mm. While typically larger than the lateral dimensions, these axial lengths still represent effective energy concentration. Focusing gain was assessed by calculating the Lateral Peak-to-Global-Average Ratio (Lateral PAR) for each focal spot, defined as the ratio of the lateral peak pressure to the average pressure within the scanned range. Experimental measurements showed Lateral PARs ranging from 7.82 dB to 10.31 dB for ASFL focal spots and 8.65 dB to 14.07 dB for OFL focal spots, across both 10 mm and 30 mm depths. These quantitative results confirm that our planar acoustic lenses effectively concentrate acoustic energy in the predefined focal regions, achieving significant pressure gain essential for effective target stimulation.

To quantitatively evaluate the side lobe suppression performance of the two lens structures in the XY plane, Figure 7a,b, respectively, present the simulated acoustic field distributions for the OFL and ASFL. To visualize the side lobe morphology more clearly, superimposed contours depict the acoustic field boundaries where the pressure amplitude drops to 50% (−6 dB) of the local peak value within each region. By calculating the areas enclosed by these contour lines, we obtained the total main lobe area, total side lobe area, and main-lobe-to-side-lobe-area ratio for the entire XY plane.

Results show that for the OFL, the total main lobe area is 28.78 mm^2^ and the total side lobe area is 4.44 mm^2^, yielding a main-lobe-to-side-lobe-area ratio of approximately 6.48. In contrast, the ASFL has a total main lobe area of 33.30 mm^2^, slightly larger than that of the OFL. However, its total side lobe area is 5.91 mm^2^, significantly larger than the OFL’s 4.44 mm^2^. Further analysis of the main-lobe-to-side-lobe-area ratio shows that the ASFL’s ratio is approximately 5.63, lower than the OFL’s 6.48. These results indicate that although the ASFL achieves a slightly larger main lobe area, its overall side lobe energy leakage is more severe. This reflects that the ASFL is less effective than the OFL in suppressing overall side lobes. This highlights the crucial role of Fresnel zone integrity in achieving effective side lobe suppression.

In applications demanding high focusing precision, such as TcFUS, undesired side lobe energy deposition is a primary cause of off-target effects [31], potentially leading to unintended neuronal activation or inhibition. Especially when stimulating deep brain regions, acoustic scattering during propagation can amplify side lobe influence, complicating precise control over the stimulated volume and potentially increasing the risk of thermal damage to non-target areas. Therefore, in scenarios prioritizing minimal side lobe impact and ensuring focusing accuracy, opting for the OFL, which exhibits a higher main-lobe-to-side-lobe-area ratio, is advisable.

To quantitatively characterize the morphological differences between the focal spots produced by the 1 mm-thick OFL and ASFL structures, post-processing analysis was performed on the acoustic field simulation results. Specifically, for each target focal region, all spatial data points where the acoustic pressure was greater than or equal to −3 dB relative to the regional maximum pressure were extracted. Subsequently, an ellipsoid was fitted to this point cloud using a linear least-squares method based on minimizing the algebraic distance, enabling precise quantification of the focal spot’s geometric features. This ellipsoid fitting algorithm first computes the centroid c of the point cloud and translates the data points $p_{i}={\left[x_{i},y_{i},z_{i}\right]}^{T}$ relative to this center to obtain centered coordinates $p_{i}^{\prime }$. It then assumes these centered points approximately satisfy the equation of a quadric surface centered at the origin:(4)$$Ax^{\prime 2}+By^{\prime 2}+Cz^{\prime 2}+2Dx^{\prime }y^{\prime }+2Ex^{\prime }z^{\prime }+2Fy^{\prime }z^{\prime }=1,$$
where A~F are the parameters to be determined respectively.

Substituting all N points into Equation (4) yields an overdetermined linear system for the algebraic coefficient vector $k={\left[A,B,C,D,E,F\right]}^{T}$,(5)$$D_{mat}k\;\approx \;1,$$
where the *i*-th row of the design matrix $D_{mat}$ consists of the quadratic terms of the centered point coordinates $p_{i}^{\prime }$, specifically $\left[x_{i}^{\prime 2},y_{i}^{\prime 2},z_{i}^{\prime 2},2x_{i}^{\prime }y_{i}^{\prime },2x_{i}^{\prime }z_{i}^{\prime },2y_{i}^{\prime }z_{i}^{\prime }\right]$. To ensure robustness in the numerical solution, the Moore–Penrose pseudoinverse is used to calculate the least-squares solution for this system,(6)$$k_{sol}={\left(D_{mat}\right)}^{†}1,$$
where 1 is a column vector of ones. The optimal coefficients $k_{sol}$ are then used to reconstruct the symmetric quadratic form matrix,(7)$$S=\left[\begin{array}{ccc} A & D & E\\ D & B & F\\ E & F & C \end{array}\right],$$
and eigenvalue decomposition is performed on S,(8)$$S=V\Lambda V^{T},$$
where $\Lambda =diag\left(\lambda_{1},\lambda_{2},\lambda_{3}\right)$ is the diagonal matrix of eigenvalues, and $V=\left[v_{1},v_{2},v_{3}\right]$ is the orthogonal matrix of corresponding eigenvectors. If all eigenvalues $\lambda_{i}>0$, the fitted surface is an ellipsoid. Its three semi-axes lengths are given by $r_{i}=\frac{1}{\sqrt{\lambda_{i}}}$, and its spatial orientation is described by the orthogonal eigenvector matrix V.

Figure 8a,b display the −3 dB point cloud distributions and corresponding fitted ellipsoids for the OFL and ASFL focal spots, respectively. Overall, the focal spots generated by both structures exhibit good left–right symmetry. However, significant morphological differences are evident between the structures, as well as between the near-field and far-field foci within the same structure. For the OFL structure, the near-field foci exhibit average semi-axes lengths along the x, y, and z directions of 1.23, 0.34, and 0.30 mm, resulting in a relatively compact morphology with an aspect ratio of 3.62. In contrast, the far-field foci demonstrate significantly larger average semi-axes lengths of 2.43, 0.40, and 0.38 mm. This reflects an expansion in all three dimensions, particularly a pronounced elongation along the x-axis, yielding a highly elongated far-field spot with an aspect ratio reaching 6.1.

In contrast, the ASFL’s near-field foci exhibited average semi-axes lengths of 2.17, 0.44, and 0.28 mm, presenting an intrinsically more elongated shape with an aspect ratio of 5.0. Its far-field foci averaged semi-axes lengths of 1.73, 0.62, and 0.34 mm. It had a shorter primary axis than its near-field counterpart, and its aspect ratio of 2.8 indicates a morphology relatively less elongated compared to the OFL’s far-field focus.

The ASFL tends to concentrate more energy into the near-field foci, evidenced by both higher peak pressures and the larger primary axis dimension of these spots. Concurrently, the ASFL maintains relative balance and consistency in the size ratios between its near-field and far-field foci. Conversely, while the OFL features slightly lower peak pressure and a more compact morphology in its near-field foci, its far-field foci display pronounced axial elongation.

These differences in morphology and energy distribution stem from varying changes in the effective acoustic transmission area within the overlapping regions of the constituent lens units. In OFL, the geometric overlap and consequent reduction in effective acoustic transmission area for the Inner FZP are significantly greater than for the Outer FZP. This leads to substantially higher acoustic energy attenuation for the near-field foci (primarily from the Inner FZP) compared to the far-field foci (primarily from the Outer FZP). The ASFL’s optimized design, however, results in more comparable reductions in transmission performance due to overlap for both inner and outer zones. Consequently, its near-field and far-field foci exhibit more balanced energy distribution and morphological proportions.

These significant differences in focal spot morphology suggest distinct potential advantages for OFL and ASFL structures in practical applications. The highly elongated far-field foci generated by the OFL are suitable for applications requiring extended axial coverage [32], while its relatively compact near-field foci can be employed for precise focusing in specific superficial regions [33]. Conversely, the ASFL, characterized by its balanced energy distribution and morphological proportions between near- and far-field foci, is better suited for applications involving simultaneous dual-depth stimulation or those demanding higher consistency in focal spot shape. The quantitative focal spot parameters derived from ellipsoid fitting in this study provide a crucial experimental and theoretical basis for optimizing acoustic lens design and predicting performance tailored to specific application requirements.

Addressing skull-induced acoustic aberration and achieving precise focusing are critical technical challenges in TcFUS. Current technical approaches primarily utilize binary acoustic metasurfaces (BAMs) [24] and complex ultrasound phased arrays [34]. Table 3 compares the technical parameters of the OFL and ASFL developed in this study with these existing methods.

Operating at 1 MHz, OFL and ASFL achieve spatial resolutions comparable to those of BAM and the 64-channel phased array system, indicating their good acoustic field focusing capability. Fabricated using 3D printing and spin coating with a composite structure of TC4 and PDMS, OFL and ASFL offer potentially superior biocompatibility and mechanical stability compared to the PLA used in BAM. Furthermore, this manufacturing approach circumvents the high electronic complexity required for the 64-channel independent drive of phased array systems, significantly reducing system cost and manufacturing complexity. Collectively, by maintaining high spatial resolution through innovative acoustic structure design, OFL and ASFL optimize system complexity and manufacturing cost, presenting a new technical path for ultrasound neuromodulation.

## 4. Conclusions

In this work, we successfully addressed three critical challenges facing TcFUS: skull-induced acoustic aberration, precise focal targeting, and the clinical need for simultaneous multi-target stimulation. To overcome these obstacles, we innovatively developed two novel planar acoustic lens structures: OFL and ASFL. By utilizing a biocompatible TC4-PDMS composite material as a skull mimic, we effectively bypassed the ultrasound phase distortion and thermal damage typically caused by the native skull. Furthermore, leveraging the favorable acoustic impedance matching between PDMS and water ensured minimal reflection loss at the interface, resulting in high-efficiency energy transmission. Through detailed characterization of the focal spot dimensions and sidelobe structure, we confirmed that the designed lenses enable precise focused stimulation with high spatial resolution. Validated by numerical simulations and underwater acoustic field measurements, we further demonstrated that these two structures can accurately generate four non-coaxial foci within a three-dimensional volume. This design principle is versatile and can be further extended to create a larger number of non-coaxial foci in both ultrasound and potentially optical modalities. This technical approach enables synergistic stimulation of multiple brain regions using a single transducer, thereby circumventing the high complexity, cost, and thermal risks associated with conventional phased array systems, while also overcoming the mechanical reliability issues of existing curved acoustic lenses. Consequently, our proposed planar lens technology offers a promising novel non-invasive solution for the therapeutic intervention of neurological diseases like Alzheimer’s and Parkinson’s, which often necessitate targeted stimulation of multiple brain regions.

However, our study also revealed inherent limitations in the current design and highlighted key directions for future improvement. Firstly, the focal spot resolution of the lens is dependent on fabrication accuracy, particularly the precision of the zone boundary dimensions. Future research could explore higher-precision fabrication techniques or error compensation strategies. Secondly, both ASFL and OFL structures exhibited suboptimal side lobe suppression performance, characterized by a less-than-ideal main-lobe-to-side-lobe-area ratio. This necessitates further optimization of zone arrangement algorithms to achieve stronger side lobe suppression. Finally, conducting biological experiments is necessary to validate the performance of these composite acoustic lenses in biological tissue environments. Overcoming these challenges is crucial for further advancing the precision, safety, and overall applicability of multi-focal TcFUS applications leveraging this promising planar lens technology.

## Figures and Tables

**Figure 1 sensors-25-03299-f001:**
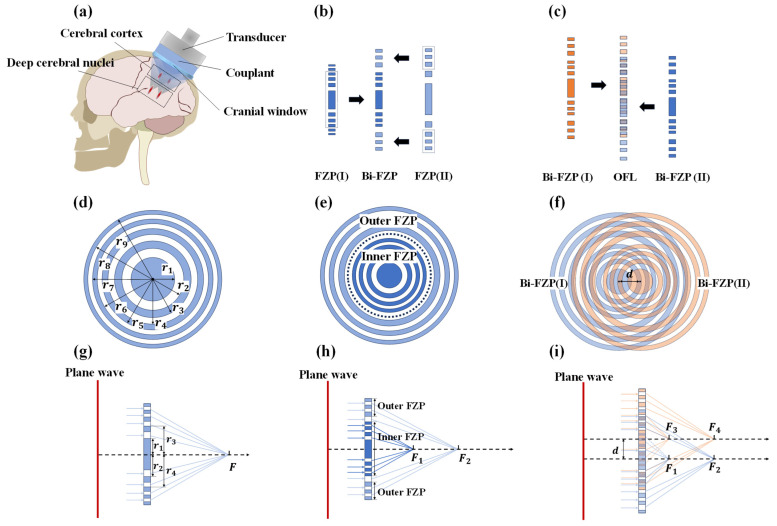
Design schematics for composite Fresnel acoustic lenses. (**a**) Schematic illustrating the application of quad-focal TcFUS; (**b**) design of the coaxial Bi-FZP; (**c**) design of the OFL; (**d**) schematic of a single-focus FZP; (**e**) structure of the Bi-FZP; (**f**) structure of the OFL; (**g**) focusing principle of the FZP; (**h**) focusing principle of the Bi-FZP; (**i**) focusing principle of the OFL.

**Figure 2 sensors-25-03299-f002:**
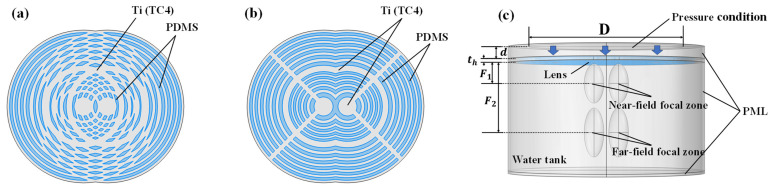
Simulation models for the designed composite Fresnel acoustic lenses. (**a**) OFL and (**b**) ASFL model; (**c**) 3D simulation model setup.

**Figure 3 sensors-25-03299-f003:**
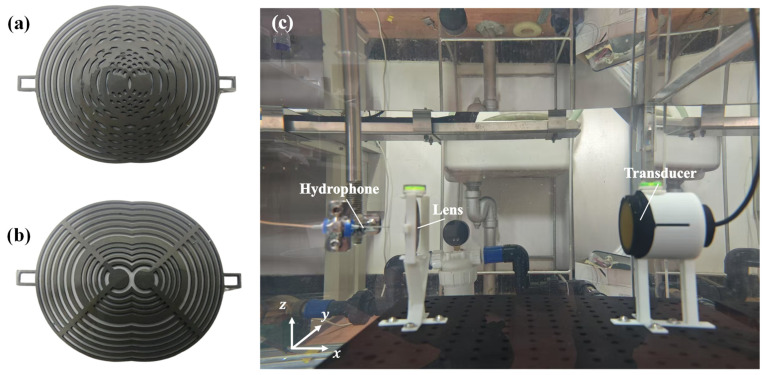
Fabricated composite Fresnel lenses and experimental setup. Fabricated (**a**) OFL and (**b**) ASFL; (**c**) experimental setup configuration.

**Figure 4 sensors-25-03299-f004:**
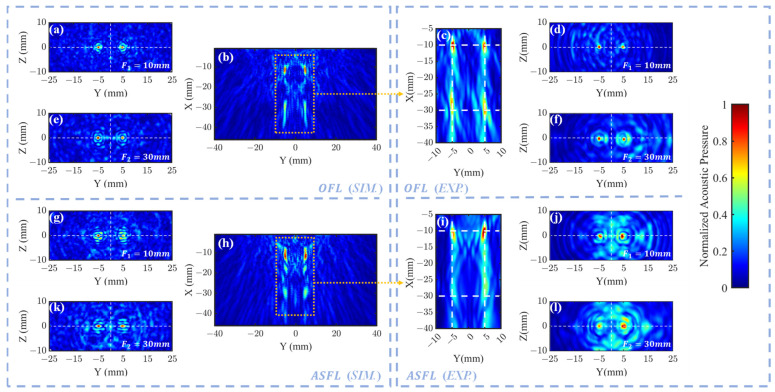
Normalized acoustic pressure distributions for 1 mm-thick OFL and ASFL. OFL: (**b**) simulated XY pressure; simulated YZ pressure at focal depths (**a**) $F_{1}=10\;\mathrm{mm}$ and (**e**) $F_{2}=30\;\mathrm{mm}$. (**c**) Experimental XY pressure; experimental YZ pressure at focal depths (**d**) $F_{1}=10\;\mathrm{mm}$ and (**f**) $F_{2}=30\;\mathrm{mm}$. ASFL: (**h**) simulated XY pressure; simulated YZ pressure at focal depths (**g**) $F_{1}=10\;\mathrm{mm}$ and (**k**) $F_{2}=30\;\mathrm{mm}$. (**i**) Experimental XY pressure; experimental YZ pressure at focal depths (**j**) $F_{1}=10\;\mathrm{mm}$ and (**l**) $F_{2}=30\;\mathrm{mm}$.

**Figure 5 sensors-25-03299-f005:**
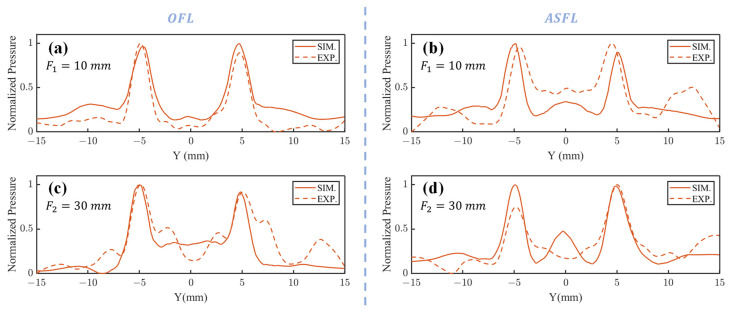
Acoustic pressure profiles along the focal axis for 1 mm-thick OFL and ASFL. OFL profiles along the focal line in the YZ plane at (**a**) $F_{1}=10\;\mathrm{mm}$ and (**c**) $F_{2}=30\;\mathrm{mm}$; ASFL profiles along the focal line in the YZ plane at (**b**) $F_{1}=10\;\mathrm{mm}$ and (**d**) $F_{2}=30\;\mathrm{mm}$.

**Figure 6 sensors-25-03299-f006:**
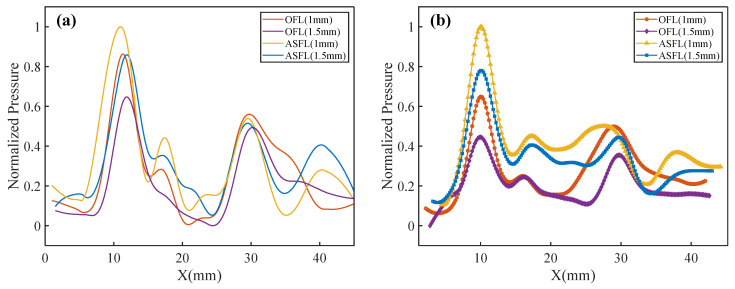
Acoustic pressure profiles along the X-axis. (**a**) Simulated and (**b**) experimental.

**Figure 7 sensors-25-03299-f007:**
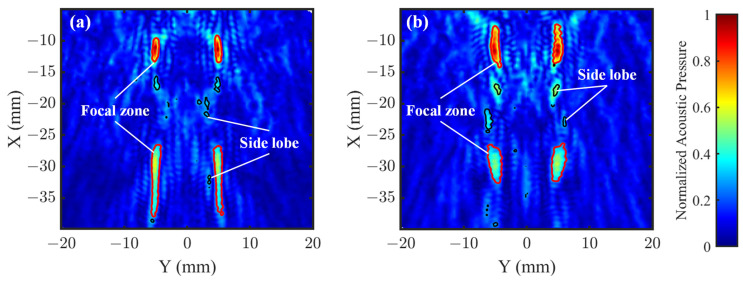
Comparison of simulated acoustic field distribution characteristics in the XY plane of (**a**) OFL and (**b**) ASFL. The contours represent the acoustic field boundaries where the acoustic pressure amplitude is 50% (−6 dB) of the local peak value. Red contours delineate the main lobe area within each localized peak region, while black contours represent the side lobe areas.

**Figure 8 sensors-25-03299-f008:**
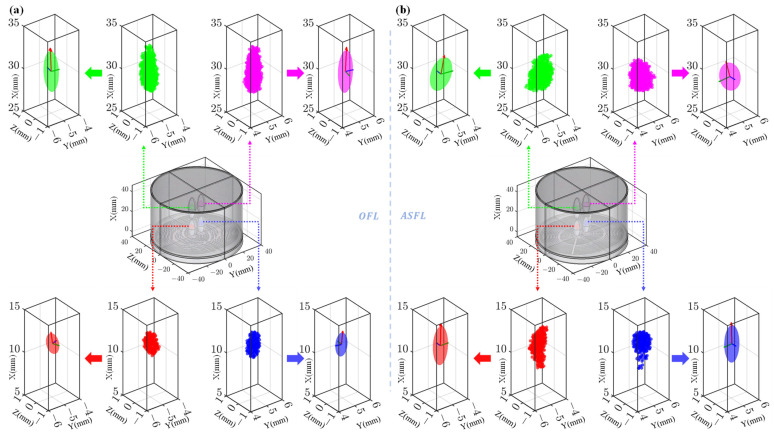
Focal spot fitting results for different lens structures. (**a**) Point clouds (−3 dB pressure boundary) and fitted ellipsoids for OFL. (**b**) Point clouds (−3 dB pressure boundary) and fitted ellipsoids for ASFL.

**Table 1 sensors-25-03299-t001:** The designed Fresnel band number corresponds to the radius.

Fresnel Zone Number ($F_{1}=10\;mm$)	$r_{n}$ (mm)	Fresnel Zone Number ($F_{2}=30\;mm$)	$r_{n}$ (mm)
1	4.00	1	6.84
2	5.76	2	9.74
3	7.18	3	12.00
4	8.43	4	13.94
5	9.58	5	15.68
6	10.67	6	17.28
7	11.70	7	18.77
8	12.69	8	20.19
9	13.66	9	21.54
10	14.60	10	22.83
11	15.53	11	24.08
12	16.44	12	25.29
13	17.33	13	26.47
14	18.22	14	27.62
15	19.09	15	28.75
16	19.95	16	29.85
17	20.81	17	30.93
18	21.66	18	32.00

**Table 2 sensors-25-03299-t002:** Acoustic material properties.

Acoustic Material	Density (kg/m^3^)	Sound Velocity (m/s)
Water	1000	1483
PDMS [29]	970	1080
Titanium alloy (TC4) [30]	4510	6100

**Table 3 sensors-25-03299-t003:** Parameter comparison between OFL/ASFL and existing approaches.

Parameter	OFL	ASFL	BAMs [24]	Ultrasound Phased Arrays [34]
Number of focal points	4	4	1 or 2	Single (enables multi-target stimulation)
Acoustic pressure FWHM	2.27 mm laterally and 7.33 mm axially at 1 MHz	2.20 mm laterally and 10.02 mm axially at 1 MHz	3.3 mm laterally and 13.5 mm axially at 0.5 MHz	1.69 mm laterally and 9.28 mm axially at 1 MHz
Manufacturing method	3D print and spin coating	3D print and spin coating	3D print	Commercial array; Clip-on holder and baseplate were 3-D printed
Material	TC4 and PDMS	TC4 and PDMS	Polylactic Acid (PLA)	Anodized aluminum housing with epoxy coating
Need for patient-specific imaging	Not Compatible	Not Compatible	CT-based skull modeling	CT scans for acoustic simulations
Focal depth	10 mm and 30 mm	10 mm and 30 mm	47.1 mm to 64.3 mm	8 mm to 18 mm
Electronic complexity	Low (single driving channel)	Low (single driving channel)	Low (single driving channel)	High (64-channel independent drive)

## Data Availability

The data underlying the results presented in this paper may be obtained from the authors upon reasonable request.

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
