# Peer review of "Design and Structure of a Non-Coaxial Multi-Focal Composite Fresnel Acoustic Lens for Synergistic Ultrasound Stimulation of Multiple Brain Regions"

_sensors, 2025, doi:10.3390/s25113299_

Round 1
Reviewer 1 Report
Comments and Suggestions for Authors
- The authors provide the construction principle and method of OFL and ASFL in the paper, but the description of the specific geometrical parameters (radius, details of the offset during superposition), etc. is not detailed enough, and it is suggested to improve it.
- The authors need to add detailed parameters of the transducer driving conditions, including input power or voltage amplitude, waveform type, transducer impedance, etc.
- The authors need to characterize the sidelobe metrics. Quantify sidelobe impact and discuss non-target stimulation risks in TcFUS applications.
- The conclusion summarizes the performance of OFL and ASFL, but does not adequately discuss design limitations (e.g., sidelobe issues, manufacturing accuracy dependence) or directions for future improvement, and additional clarification is recommended.
- Inconsistent use of terms like “OFL” and “overlapping Fresnel lens.” Standardize terminology throughout the manuscript.
Reviewer 2 Report
Comments and Suggestions for Authors
This article discusses the impediment of transcranial focused ultrasound neuromodulation caused by acoustic limitations from the skull. To achieve synergistic multi-regional brain stimulation, the authors designed non-coaxial multi-focal composite Fresnel acoustic lenses: overlapping Fresnel lenses and alternately segmented Fresnel lenses. These lenses convert planar ultrasound into multiple focal points. Based on Fresnel theory, this design approach has the potential to extend to other non-coaxial ultrasound and even optical focusing. By facilitating synergistic multi-regional stimulation using a single sensor, this method avoids the complexity and thermal risks of phased array systems while overcoming the mechanical reliability challenges of curved acoustic lenses. Therefore, it provides a new non-invasive solution for treating neurological diseases that require multi-target intervention strategies, such as Alzheimer's disease and Parkinson's disease.
Q1: In Section 2, Figure 1 (line 88) sequentially introduces figures a-j, it is suggested to add h in order.
Q2: In Section 2, Figure 3 (line 166), it is suggested to remove the white background text in Figure (c) for better aesthetics.
Q3: In Section 2, Figure 5 (line 202), it is suggested to arrange the figure horizontally as in the previous figure (Figure 1).
Q4: The article introduces the experimental setup but does not provide complete experimental content. It is recommended to elaborate on the experimental setup and operational procedures, ensuring the repeatability of the experiments.
Q5: The article mainly analyzes the effects on lenses under different parameters. It is suggested to include some comparative analysis with previous research in this area.
Reviewer 3 Report
Comments and Suggestions for Authors
The manuscript convincingly demonstrates that a planar composite Fresnel lens can create four off‑axis foci for transcranial neuromodulation with a single 1 MHz transducer. Both the experimental protocol and the agreement between measured and simulated pressure fields are clearly presented and technically sound.
That said, the narrative would benefit from stronger alignment between the Introduction and the later sections. The opening paragraphs raise three challenges—skull‑induced aberration, focal precision, and the clinical need for multi‑target stimulation—but the Results and Discussion do not revisit them in detail. Please ensure that every problem posed early on is subsequently addressed, or else mark it explicitly as work for future studies. For instance, the Introduction notes that phased arrays can correct skull‑related phase errors; the manuscript should explain whether, and how, the proposed lens mitigates the same issue.
The term “precise” is used repeatedly but never quantified. Define a clinically meaningful accuracy metric—such as a maximum focal‑offset tolerance of ≤1 mm or an allowable peak‑pressure drop of ≤10 %—and demonstrate that the reported data meet this threshold.
Finally, a benchmark table contrasting the proposed lens with representative phased‑array implementations would substantiate the claimed practical advantages. Suggested columns might include acoustic/electrical power, fabrication cost, focal accuracy, need for patient‑specific calibration or imaging, and electronic complexity (number of driven channels).
Addressing these points will tighten the manuscript’s focus, quantify its key claims, and help readers weigh the proposed approach against state‑of‑the‑art alternatives.
